# Beyond Uniformity: Sample and Frequency Meta Weighting for Post-Training Quantization of Diffusion Models

**Cuong Pham**[1]  **Hoang Anh Dung**[1]  **Cuong C. Nguyen**[2]  **Trung Le**[1]
**Dinh Phung**[1]  **Gustavo Carneiro**[2]  **Thanh-Toan Do**[1]

[1]Department of Data Science and AI, Monash University, Australia
[2]Centre for Vision, Speech and Signal Processing, University of Surrey, United Kingdom
[1]{cuong.pham1, hoang.dung, trunglm, dinh.phung, toan.do}@monash.edu
[2]{c.nguyen, g.carneiro}@surrey.ac.uk

## Abstract

Post-training quantization (PTQ) is an attractive approach for compressing diffusion models to speed up the sampling process and reduce memory footprint. Most existing PTQ methods uniformly sample data from various time steps in denoising process to construct a calibration set for quantization and consider calibration samples equally important during the quantization process. However, treating all calibration samples equally may not be optimal. One notable property in the denoising process of diffusion models is that low-frequency features are primarily recovered in early stages, while high-frequency features are recovered in later stages of the denoising process. However, none of the previous works on quantization for diffusion models consider this property to enhance the effectiveness of quantized models. In this paper, we propose a novel meta-learning approach for PTQ of diffusion models that jointly optimizes the contributions of calibration samples and the weighting of frequency components at each time step for quantizing noise estimation networks. Specifically, our approach automatically learns to assign optimal weights to calibration samples while selectively focusing on mimicking specific frequency components of data generated by the full-precision noise estimation network at each denoising time step. Extensive experiments on CIFAR-10, LSUN-Bedrooms, FFHQ, and ImageNet datasets demonstrate that our approach consistently outperforms the compared PTQ methods for diffusion models.

## 1 Introduction

Recently, diffusion models (Ho et al., 2020; Dhariwal & Nichol, 2021; Rombach et al., 2022) have attracted significant attention due to their ability to generate high-quality images. However, the sampling process in diffusion models is computationally expensive, requiring hundreds of denoising steps to generate a high-quality image. Additionally, the noise estimation networks in diffusion models are often complex and have a large number of parameters, which limits diffusion models' practical applications on resource-constrained devices. To address these challenges, an attractive approach is to quantize diffusion models. Neural network quantization (Han et al., 2016; Courbariaux et al., 2015; Nagel et al., 2019; 2020; Cai et al., 2020) is a popular approach for model compression that can significantly reduce computational cost and memory usage. Post-training quantization (PTQ) is a particularly effective quantization approach due to its ability to quantize deep neural networks without relying on a large amount of training data or necessitating model retraining.

Calibration data plays a crucial role in PTQ for diffusion models and is typically generated from various time steps of the denoising process. There are several works that use heuristics to select calibration data for PTQ on diffusion models. For example, in PTQ4DM (Shang et al., 2023), the authors sample denoising time steps from a distribution $\mathcal{N}(\mu, 0.5T)$ where $\mu \leq 0.5T$, and use images generated at these sampled time steps as calibration data. In Q-Diffusion (Li et al., 2023), the

authors select generated images at fixed step intervals across all denoising time steps as calibration data. In TFMQ-DM (Huang et al., 2024), the authors adopt the Q-Diffusion method to construct the calibration data and propose a temporal feature maintenance quantization framework to improve the performance of the PTQ for diffusion models. It is worth noting that in previous works (Shang et al., 2023; Li et al., 2023; Huang et al., 2024), calibration samples are treated equally during the quantization process. Different from previous works (Shang et al., 2023; Li et al., 2023; Huang et al., 2024), we hypothesize that each calibration sample could have different contributions to the performance of the quantized model. To validate this, we conduct an empirical study by comparing uniform sample weighting against multiple random weighting schemes on the CIFAR-10 dataset. As shown in Figure 1, among 50 different weighting schemes, 18 outperform uniform weighting in terms of FID score, demonstrating that uniform weighting is suboptimal and that better weighting solutions exist. Therefore, unlike previous methods that treat calibration samples equally, we propose a principled approach inspired by (Ren et al., 2018) to automatically weight the contribution of each calibration sample for quantizing the noise estimation network. Specifically, we propose a sample-weighting mechanism that leverages meta-learning to automatically learn a weight for each calibration sample, with the objective that the quantized model trained with the calibration samples and their corresponding weights can achieve good performance on the validation set.

Furthermore, previous works (Yang et al., 2023; Qian et al., 2024) show that each time step in the diffusion process learns distinct features and serves a unique role in the diffusion model. From a frequency perspective, diffusion models recover low-frequency features at early denoising stages and gradually add high-frequency features at the later denoising stages (Qian et al., 2024). Therefore, the quantization of the noise estimation network should focus on different frequency components at different time steps during the quantization process. To this end, we propose a novel approach for PTQ for diffusion models by utilizing these frequency characteristics. Specifically, we propose a frequency weighting method that leverages meta-learning to automatically assign weights to the frequency loss components, derived from the frequencies of features extracted by the full-precision and quantized noise estimation networks at each time step, such that the learned frequency weights lead to the minimization of the quantized model's validation loss. Additionally, we propose a regularization term on frequency weights to encourage the quantized

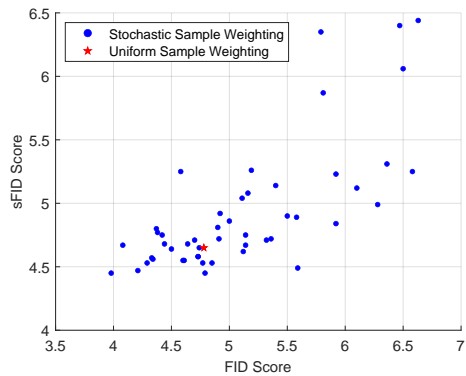

Figure 1: Comparison of FID and sFID scores of 50 sample weighting sets including the uniform weighting in the quantized noise estimation network DDPM (Ho et al., 2020) with the W4A32 setting on the CIFAR-10 dataset. All experiments use the same calibration dataset generated from the full-precision model and the same approach with TFMQ (Huang et al., 2024).

model to focus more on mimicking high-frequency components and pay less attention to low-frequency components of samples generated by the full-precision model as the time step decreases during the denoising process.

We form the optimization of the sample weights and frequency weights as a bi-level optimization problem. The aim of the optimization is to learn sample weights and frequency weights such that the quantized model obtained from the training using calibration samples with those weights achieve a good performance on the validation set, i.e., minimizing the validation loss. We validate our proposed approach on the widely used CIFAR-10 (Krizhevsky et al., 2010), LSUN-Bedrooms (Yu et al., 2015), FFHQ (Karras et al., 2019), and ImageNet (Deng et al., 2009) datasets with various noise estimation network architectures under different bit-width settings. The extensive experiments demonstrate that our method outperforms compared PTQ methods for diffusion models. To summarize, the contributions of this paper are outlined as follows:

- We propose a novel PTQ method that leverages meta-learning to automatically learn to weight the contribution of each calibration sample in PTQ training for diffusion models.

Such a weighting mechanism prioritizes important samples, improving the performance of the quantized model.

- We propose a meta-learning based method to automatically learn to weight components of the frequency loss. We also propose a regularization term to encourage the quantized model to focus more on mimicking high-frequency components and pay less attention to low-frequency components of the data generated from the full-precision counterpart as the time step decreases during the denoising process.

- We extensively validate our proposed approach on the CIFAR-10, LSUN-Bedrooms, FFHQ, and ImageNet datasets. The experimental results show that our method consistently outperforms compared PTQ methods for diffusion models in terms of the FID score.

## 2 RELATED WORKS

**Post-training quantization of diffusion models.** Diffusion models (Ho et al., 2020; Song et al., 2021b) can generate high-quality images through an iterative denoising process. However, the excessive cost of a large number of time steps in the denoising process could limit the practical applications of diffusion models. Although several works significantly reduce sampling time (Lu et al., 2022; Song et al., 2021a; Zhao et al., 2023), they still face challenges in computational cost and memory usage due to complex noise estimation networks. Model compression, especially model quantization (Han et al., 2016; Courbariaux et al., 2015; Nagel et al., 2019; 2020; Cai et al., 2020; Xu et al., 2020), is an effective approach to reduce the computational cost and memory usage of these networks. Post-training quantization (Nagel et al., 2020; Li et al., 2021; Liu et al., 2023; Wei et al., 2022; Jeon et al., 2023) is an effective approach to quantize diffusion models. This family of techniques requires constructing appropriate calibration data and a quantization scheme for the model quantization. Existing PTQ methods for diffusion models mainly focus on obtaining calibration samples. To construct the calibration data, PTQ4DM (Shang et al., 2023) shows that generated samples in the denoising process are better than those from the forward process for PTQ for diffusion models. Q-Diffusion (Li et al., 2023) improves upon this by selecting generated images at fixed step intervals across all denoising time steps and introducing shortcut-splitting quantization, achieving enhanced performance across a broader dataset range. In APQ-DM (Wang et al., 2024), the authors propose using the structural risk minimization principle to find optimal time steps for generating calibration data. However, these works (Shang et al., 2023; Li et al., 2023; Wang et al., 2024) treat all calibration samples with equal importance during the quantization process, ignoring the fact that certain samples may contribute more critically to model performance than others.

**Frequency in diffusion models.** Frequency information has been widely adopted in conventional generative models, such as GANs (Fu et al., 2021; Yang et al., 2022; Zhang et al., 2022). Recently, several works have leveraged the frequency domain information to improve the performance of diffusion models (Yang et al., 2023; Phung et al., 2023; Qian et al., 2024). In (Phung et al., 2023), the authors propose frequency-aware architectures for diffusion models to reduce the inference time while maintaining high quality of generated samples. From a temporal perspective, different time steps in the diffusion process exhibit distinct frequency characteristics (Yang et al., 2023). The denoising process typically recovers low-frequency features in early time steps before gradually incorporating high-frequency details in later stages (Yang et al., 2023). Spectral Diffusion (Yang et al., 2023) exploits this frequency evolution through wavelet gating for spectrum-aware distillation. In (Qian et al., 2024), the authors propose a training-free approach that leverages frequency domain information to enhance the stability of the denoising process and improve the performance of diffusion models. While there are previous works exploiting the frequency domain information to improve the performance of full-precision diffusion models, research on leveraging frequency domain information for the quantization of diffusion models to improve quantized diffusion model performance remains limited.

**Meta-learning for post-training quantization.** Meta-learning has been explored for convolutional neural network quantization (Chen et al., 2019; Wang et al., 2020; Youn et al., 2022; Kim et al., 2024; Pham et al., 2024). For instance, MEBQAT (Youn et al., 2022) leverages meta-learning to optimize a mixed-precision quantization strategy that swiftly adapts to diverse bit-width configurations while preserving model accuracy. In another approach, MetaMix (Kim et al., 2024) addresses the prevalent

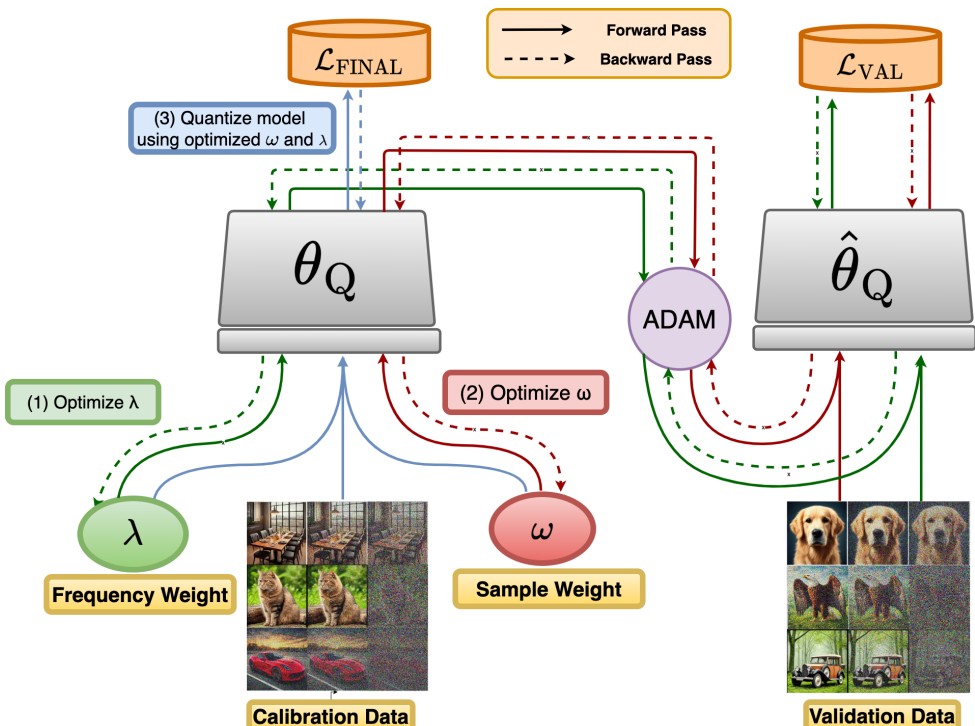

Figure 2: In general, the proposed method consists of three main optimization steps: (1) updating frequency weight $\lambda$, (2) updating sample weight $\omega$ using the validation loss $\mathcal{L}_{\text{val}}$ in Eq. (7), and (3) leveraging both sample and frequency weights to quantize the model by minimizing the final loss $\mathcal{L}_{\text{FINAL}}$ in Eq. (12).

issue of activation instability in mixed-precision quantization models and utilizes meta-learning to mitigate this instability and improve robustness. On the other hand, MetaQuantNet (Wang et al., 2020) presents a meta-learning framework that autonomously identifies optimal quantization policies before employing these policies to enhance network quantization. However, the use of meta-learning in diffusion quantization settings remains largely unexplored. To the best of our knowledge, our work is the first to leverage meta-learning techniques specifically to post-training quantization within the context of diffusion models.

## 3 PROPOSED METHOD

### 3.1 PRELIMINARY DEFINITIONS

Our goal is to optimize a set of sample weights and frequency weights that guide the quantization process to better preserve the full-precision model's behavior. The details of the algorithm are illustrated in Figure 2. We first obtain the calibration set following the approach outlined in the Q-Diffusion method (Li et al., 2023), by selecting generated samples at fixed step intervals across all denoising time steps. Each calibration sample is denoted as $(x_i, t_i)$, where $x_i$ is the generated sample with the corresponding time step $t_i$.

In our method, each calibration sample $(x_i, t_i)$ is assigned a learnable weight $\omega_i$, representing its impact on the quantized model's performance. The set of $N$ weights corresponding to $N$ calibration samples is denoted as $\omega = \{\omega_i\}_{i=1}^{N}$.

Besides the sample weighting, we propose to weight individual frequency components for each time step to better align with the evolution dynamics of different frequency components. Specifically, the Discrete Wavelet Transform (DWT) (Graps, 1995) is a well-known frequency analysis method. In this paper, we utilize DWT and leverage the frequency properties of diffusion models for quantization, thereby enhancing the effectiveness of quantized models. In practice, any tensor input $\boldsymbol{a}$ is

decomposed into four wavelet subbands by applying DWT (Graps, 1995) as follows:

$$\text{DWT}(\boldsymbol{a}) = (\boldsymbol{a}_{ll}, \boldsymbol{a}_{lh}, \boldsymbol{a}_{hl}, \boldsymbol{a}_{hh}). \tag{1}$$

Note that here we implement DWT as the classical Haar wavelet (Stankovic & Falkowski, 2003) for simplicity. Among the four wavelet subbands, $\boldsymbol{a}_{ll}$ refers to the low-frequency component that reflects the basic object structure, while $\boldsymbol{a}_{\{lh,hl,hh\}}$ represent high-frequency components that capture texture details. When quantizing the $l^{th}$ layer of the network, we assign all calibration samples $x_i$ of time step $t_i$ with a set of learnable weights $\{\lambda_{t_i,0}, \lambda_{t_i,1}, \lambda_{t_i,2}, \lambda_{t_i,3}\}$, denoting the weights corresponding to $\boldsymbol{a}_{ll}, \boldsymbol{a}_{lh}, \boldsymbol{a}_{hl}, \boldsymbol{a}_{hh}$ at time step $t_i$. Let us denote $\lambda$ as a learnable frequency weight matrix of size $T \times 4$, where the $t_i^{th}$ row $\lambda_{t_i} = \{\lambda_{t_i,0}, \lambda_{t_i,1}, \lambda_{t_i,2}, \lambda_{t_i,3}\}$ is a vector of length 4. We normalize the total weight of all frequency components at each time step $t_i$ equal to 1 (i.e. $\sum_{k=0}^{3} \lambda_{t_i,k} = 1$).

## 3.2 THE JOINT OPTIMIZATION OF SAMPLE AND FREQUENCY WEIGHTS

Both the sample weight $\omega$ and the frequency weight $\lambda$ are optimized to maximize the model's performance on the validation set. Given a full-precision model $\theta_{\text{FP}}$ and a quantized model $\theta_{\text{Q}}$, the joint optimization objective of $\lambda$ and $\omega$ is formed as a bi-level optimization problem as follows:

$$\omega^*, \lambda^* = \underset{\omega, \lambda}{\arg\min} \frac{1}{|S^v|} \sum_{x_j \in S^v} \mathcal{L}_{\text{val}}\left(\widehat{\theta}_{\text{Q}}, x_j, \lambda\right), \tag{2}$$

$$\text{s.t: } \widehat{\theta}_{\text{Q}} = \underset{\theta_{\text{Q}}}{\arg\min} \sum_{x_i \in S^c} \omega_i \left[\mathcal{L}_{\text{Q}}(\theta_{\text{Q}}, x_i, l) + \gamma \mathcal{L}_{\text{F}}(\theta_{\text{Q}}, x_i, \lambda, l)\right], \tag{3}$$

where $S^c$ and $S^v$ are the calibration dataset and validation dataset, respectively; $|S|$ denotes the cardinality of the set $S$; $l$ is the index of the layer/block that we want to calibrate, and $\gamma$ is a hyperparameter.

**Regarding the loss $\mathcal{L}_{\text{Q}}$ in Eq. (3).** The loss $\mathcal{L}_{\text{Q}}$ is used to update the $l^{th}$ block of the model $\theta_{\text{Q}}$ to obtain the model $\widehat{\theta}_{\text{Q}}$, which is defined as follows:

$$\mathcal{L}_{\text{Q}}(\theta_{\text{Q}}, x_i, l) = \left\|\boldsymbol{\epsilon}_{\text{FP}}^{(l)}(x_i, t_i) - \boldsymbol{\epsilon}_{\text{Q}}^{(l)}(x_i, t_i)\right\|^2, \tag{4}$$

where $\boldsymbol{\epsilon}_{\text{FP}}^{(l)}(x_i, t_i)$ and $\boldsymbol{\epsilon}_{\text{Q}}^{(l)}(x_i, t_i)$ are the outputs of the $l^{th}$ block of the full-precision model $\theta_{\text{FP}}$ and the quantized model $\theta_{\text{Q}}$ for sample $(x_i, t_i)$, respectively.

**Regarding the loss $\mathcal{L}_{\text{F}}$ in Eq. (3).** During the quantization process, we encourage the frequency components of images generated by the quantized model to match their counterparts from the full-precision model. Specifically, for any two tensors $\boldsymbol{a}, \boldsymbol{b}$ of the same size, we employ DWT to decompose two tensors $\boldsymbol{a}, \boldsymbol{b}$ into four wavelet subbands as $\text{DWT}(\boldsymbol{a}) = (\boldsymbol{a}_{ll}, \boldsymbol{a}_{lh}, \boldsymbol{a}_{hl}, \boldsymbol{a}_{hh})$ and $\text{DWT}(\boldsymbol{b}) = (\boldsymbol{b}_{ll}, \boldsymbol{b}_{lh}, \boldsymbol{b}_{hl}, \boldsymbol{b}_{hh})$. Given a frequency weight vector $\widehat{\lambda} = (\widehat{\lambda}_0, \widehat{\lambda}_1, \widehat{\lambda}_2, \widehat{\lambda}_3)$, their weighted frequency difference will be defined as:

$$\mathcal{L}_f(\boldsymbol{a}, \boldsymbol{b}, \widehat{\lambda}) = \widehat{\lambda}_0 \|\boldsymbol{a}_{ll} - \boldsymbol{b}_{ll}\|^2 + \widehat{\lambda}_1 \|\boldsymbol{a}_{lh} - \boldsymbol{b}_{lh}\|^2 + \widehat{\lambda}_2 \|\boldsymbol{a}_{hl} - \boldsymbol{b}_{hl}\|^2 + \widehat{\lambda}_3 \|\boldsymbol{a}_{hh} - \boldsymbol{b}_{hh}\|^2. \tag{5}$$

Then the frequency loss $\mathcal{L}_{\text{F}}$ can be defined as follows:

$$\mathcal{L}_{\text{F}}(\theta_{\text{Q}}, x_i, \lambda, l) = \mathcal{L}_f\left(\boldsymbol{\epsilon}_{\text{Q}}^{(l)}(x_i, t_i), \boldsymbol{\epsilon}_{\text{FP}}^{(l)}(x_i, t_i), \lambda_{t_i}\right). \tag{6}$$

**Regarding the loss $\mathcal{L}_{\text{val}}$ in Eq. (2).** Our goal is to maximize the performance of model $\widehat{\theta}_{\text{Q}}$ on the validation set $S^v$. Therefore, at the validation step, we validate the quantized model $\widehat{\theta}_{\text{Q}}$ on the validation set $S^v$. The validation loss $\mathcal{L}_{\text{val}}$ is presented as below:

$$\mathcal{L}_{\text{val}}(\widehat{\theta}_{\text{Q}}, x_j, \lambda) = \|\boldsymbol{\epsilon}_{\text{FP}}(x_j, t_j) - \boldsymbol{\epsilon}_{\text{Q}}(x_j, t_j)\|^2 + \beta \mathcal{L}_{\text{Reg}}(\lambda), \tag{7}$$

where $\boldsymbol{\epsilon}$ with a subscript is the final output of the corresponding model of interest; $\beta$ is a hyperparameter. The first term in the validation loss, $\mathcal{L}_{\text{val}}$, is the reconstruction loss between the outputs of the full-precision model and the quantized model for each sample $x_j$ in the validation dataset $S^v$, while the second term represents the regularization loss on frequency weights.

**Regarding the regularization $\mathcal{L}_{\text{Reg}}$ in Eq. (7).** As the full-precision model gradually recovers the low-frequency components at the early stages of the denoising process and gradually recovers the high-frequency components at the later stages, the quantized model is encouraged to follow this pattern. Specifically, the weight of the low-frequency components ($\lambda_{t,0}$) is regularized to be decreased and the weights of the high-frequency components ($\lambda_{t,1} + \lambda_{t,2} + \lambda_{t,3}$) in frequency loss $\mathcal{L}_{\text{F}}$ is regularized to be increased as the time step decreases. To this end, we define the regularization loss $\mathcal{L}_{\text{Reg}}$ as:

$$\mathcal{L}_{\text{Reg}}(\lambda) = \sum_{t=0}^{T-2} \max(0, \mathbf{r}_t - \mathbf{r}_{t+1}), \tag{8}$$

where $\mathbf{r} = \lambda_{:,0} \oslash (\lambda_{:,1} + \lambda_{:,2} + \lambda_{:,3})$, and $\oslash$ denotes element-wise division. As the time step $t$ decreases, the regularization loss $\mathcal{L}_{\text{Reg}}(\lambda)$ will encourage $\lambda_{t,0}$ to decrease while $\lambda_{t,1} + \lambda_{t,2} + \lambda_{t,3}$ to increase.

When optimizing the sample weights $\{\omega_i\}_{i=1}^N$, we keep the frequency weight $\lambda$ fixed throughout this step, and conversely. The model $\widehat{\theta}_Q(\omega)$, when optimizing sample weights $\omega$, is approximated by solving Eq. (3) using a single step of gradient-based methods (e.g., SGD or Adam) as follows:

$$\widehat{\theta}_{\text{Q}} = \theta_{\text{Q}} - \eta_Q \sum_{x_i \in S^c} \boldsymbol{\nabla}_{\theta_{\text{Q}}} \omega_i \left[ \mathcal{L}_{\text{Q}}(\theta_{\text{Q}}, x_i, l) + \gamma \mathcal{L}_{\text{F}}(\theta_{\text{Q}}, x_i, \lambda, l) \right], \tag{9}$$

where $\eta_Q$ denotes the learning rate of the quantized model.

In the second stage, we optimize the sample weights $\{\omega_i\}_{i=1}^N$ and the frequency weight matrix $\lambda$ with respect to the quantized model $\widehat{\theta}_{\text{Q}}$. We employ an alternating optimization scheme in which one set of parameters is held fixed while the other is updated. Concretely, the sample weights are updated as:

$$\omega_i = \omega_i - \eta_\alpha \frac{1}{|S^v|} \sum_{x_j \in S^v} \frac{\partial \mathcal{L}_{\text{val}}\left(\widehat{\theta}_{\text{Q}}(\omega), x_j, \lambda\right)}{\partial \omega_i}, \forall i = 1, 2, \ldots, N, \tag{10}$$

followed by an update of the frequency weights:

$$\lambda_{t,k} = \lambda_{t,k} - \eta_\lambda \frac{1}{|S^v|} \sum_{x_j \in S^v} \frac{\partial \mathcal{L}_{\text{val}}\left(\widehat{\theta}_{\text{Q}}(\lambda), x_j, \lambda\right)}{\partial \lambda_{t,k}}, \forall t \in \{1, \ldots, T\}, k \in \{0, 1, 2, 3\}, \tag{11}$$

where $\eta_\lambda$ denotes the learning rate of the frequency weight matrix $\lambda$, $\eta_\alpha$ denotes the learning rate of the sample weights. In the sample weight optimization step, since the frequency weight matrix $\lambda$ is fixed, $\mathcal{L}_{\text{Reg}}(\lambda)$ in Eq. (7) is ignored.

### 3.3 FINAL OPTIMIZATION OBJECTIVE

For the $l^{th}$ layer/block, once we have obtained the sample weights $\{\omega_i\}_{i=1}^N$ and the frequency weight matrix $\lambda$ corresponding to that layer/block, the model will be quantized over the training set with a combined loss, defined as:

$$\mathcal{L}_{\text{FINAL}} = \sum_{i=1}^N \omega_i [\mathcal{L}_{\text{Q}}(\theta_{\text{Q}}, x_i, l) + \gamma \mathcal{L}_{\text{F}}(\theta_{\text{Q}}, x_i, \lambda, l)]. \tag{12}$$

The overall algorithm of our proposed method is presented in Algorithm 1.

## 4 EXPERIMENTS

### 4.1 EXPERIMENTAL SETUP

**Models and datasets.** We evaluate the performance of our proposed method on common diffusion models, including the pixel-space diffusion model DDPM (Ho et al., 2020) for unconditional image

---

**Algorithm 1** Sample and frequency meta weighting for post-training quantization of diffusion models

---

1: **procedure** $\text{TRAIN}(\theta_{\text{FP}}, S^c, S^v)$
2:     ▷ *$\theta_{\text{FP}}$: full-precision model* ◁
3:     ▷ *$\theta_{\text{Q}}$: quantized model* ◁
4:     ▷ *$L$: number of blocks in the full-precision model* ◁
5:     ▷ *$S^c$: calibration dataset* ◁
6:     ▷ *$S^v$: validation dataset* ◁
7:     ▷ *$N_f$: number of iterations for updating frequency weight $\lambda$* ◁
8:     ▷ *$N_s$: number of iterations for updating sample weights $\{\omega_i\}_{i=1}^N$* ◁
9:     ▷ *$N_Q$: number of iterations for model weight quantization* ◁
10:     Uniformly initialize sample weights $\omega = \{\omega_i\}_{i=1}^N$ and frequency weight $\lambda$
11:     Initialize the quantized model $\theta_{\text{Q}}$
12:     **for** $l = 1$ to $L$ **do**
13:         **while** not converged **do**
14:             ▷ *Fix $\{\omega_i\}_{i=1}^N$ and update frequency weight $\lambda$* ◁
15:             **for** $n_f = 1$ to $N_f$ **do**
16:                 Compute $\widehat{\theta}_Q(\lambda)$ using Eq. (9)
17:                 Compute $\mathcal{L}_{\text{val}}(\widehat{\theta}_Q(\lambda), S^v, \lambda)$ using Eq. (7)
18:                 Update $\lambda$: $\lambda \leftarrow \text{ADAM}(\mathcal{L}_{\text{val}}(\widehat{\theta}_Q(\lambda), S^v, \lambda))$
19:             ▷ *Fix $\lambda$ and update sample weights $\{\omega_i\}_{i=1}^N$* ◁
20:             **for** $n_s = 1$ to $N_s$ **do**
21:                 Compute $\widehat{\theta}_Q(\omega)$ using Eq. (9)
22:                 Compute $\mathcal{L}_{\text{val}}(\widehat{\theta}_Q(\omega), S^v, \lambda)$ using Eq. (7)
23:                 Update $\{\omega_i\}_{i=1}^N$: $\{\omega_i\}_{i=1}^N \leftarrow \text{ADAM}(\mathcal{L}_{\text{val}}(\widehat{\theta}_Q(\omega), S^v, \lambda))$
24:         ▷ *Optimize parameters of the quantized model* ◁
25:         **for** $n_q = 1$ to $N_Q$ **do**
26:             Optimize the quantizer parameters of the $l^{th}$ block in model $\theta_{\text{Q}}$ by minimizing
                $\mathcal{L}_{\text{FINAL}}$ from Eq. (12) over $S^c$ using weights $\{\omega_i\}_{i=1}^N$ and $\lambda$.
27:     **return** quantized model $\theta_{\text{Q}}$

---

generation, and the latent-space diffusion model LDM (Rombach et al., 2022) for both unconditional and class-conditional image generation. We extensively evaluate the proposed method on various datasets, including CIFAR-10 $32 \times 32$ (Krizhevsky et al., 2010), LSUN-Bedrooms $256 \times 256$ (Yu et al., 2015), FFHQ $256 \times 256$ (Karras et al., 2019), and ImageNet $256 \times 256$ (Deng et al., 2009).

**Implementation details.** We follow state-of-the-art post-training quantization (PTQ) methods for both weights and activations in diffusion models (Shang et al., 2023; Huang et al., 2024). Specifically, weights and activations in PTQ for DM are typically quantized separately. We first keep the activations in full precision while quantizing the weights. For weight quantization, we learn the rounding function using AdaRound (Nagel et al., 2020) and use block-wise reconstruction (Li et al., 2021) to quantize the noise estimation networks. On the other hand, applying a similar approach to optimize activation quantizers may introduce additional training overhead while only yielding minimal performance gains, as outlined in TFMQ-DM (Huang et al., 2024). Therefore, we adopt the simpler activation quantization approach used in TFMQ-DM. This approach estimates activation ranges using EMA (Jacob et al., 2018) with a mini-batch size of 16. The quantized model $\theta_Q$ is initialized from the full-precision model using LAPQ (Nahshan et al., 2021), following previous works (Shang et al., 2023; Li et al., 2023; Huang et al., 2024). The calibration data is generated through the full-precision diffusion models as described in Q-Diffusion (Li et al., 2023) and is identical to the calibration set used in TFMQ-DM (Huang et al., 2024). We also adopt the temporal feature maintenance quantization technique in the TFMQ-DM (Huang et al., 2024) method. The number of iterations $N_Q$ for optimizing each block of the quantized model is $2 \times 10^4$ iterations following previous works (Shang et al., 2023; Li et al., 2023; Huang et al., 2024). Meanwhile, we set $N_f = 100$ and $N_s = 200$ for updating frequency weight and sample weight, respectively. We employ the Adam optimizer (Kingma & Ba, 2015) with a learning rate of $4 \times 10^{-5}$ to update the

Table 1: Quantization results for unconditional image generation with DDIM on CIFAR-10 $32 \times 32$.

| Methods | CIFAR-10 $32 \times 32$ | | | | | | | |
| --- | --- | --- | --- | --- | --- | --- | --- | --- |
| | W/A | FID↓ | sFID↓ | W/A | FID↓ | sFID↓ | W/A | FID↓ | sFID↓ |
| PTQ4DM (Shang et al., 2023) | 4/32 | 5.65 | - | 4/8 | 5.14 | - | 8/8 | 5.69 | - |
| Q-Diffusion (Li et al., 2023) | | 5.08 | 4.98 | | 4.98 | 5.68 | | 4.78 | 4.75 |
| APQ-DM (Wang et al., 2024) | | 9.96 | 7.63 | | 12.2 | 7.66 | | 6.34 | 4.44 |
| TFMQ-DM (Huang et al., 2024) | | 4.73 | - | | 4.78 | - | | 4.24 | - |
| TCAQ-DM (Huang et al., 2025) | | 4.28 | - | | 4.59 | - | | **4.09** | - |
| Ours | | **4.21** | **4.47** | | **4.25** | **4.46** | | 4.15 | **4.36** |

sample weight $\omega$ and frequency weight $\lambda$. Gradients in Eq. (10) and Eq. (11) are calculated using the *higher* library[1]. The hyper-parameter $\gamma$ is set to 0.1 in Eq. (9) and Eq. (12). When optimizing the frequency weight $\lambda$, we set the $\beta = 0.05$ for the $\mathcal{L}_{val}$ in Eq. (7). Regarding the validation set $S^v$, we use a subset of the generated data as the validation set.

The quantization settings in our proposed method are consistent with those used in Q-Diffusion (Li et al., 2023), PTQD (He et al., 2023), and TFMQ-DM (Huang et al., 2024). In line with these works, we utilize pre-trained diffusion models from the official implementations of DDIM (Song et al., 2021a) and Latent Diffusion (Rombach et al., 2022). For evaluating FID and sFID scores, we adopt the torch-fidelity library[2]. Following the setting from (Li et al., 2023; Huang et al., 2024), we use 100 denoising time steps for DDIM on the CIFAR-10 dataset. For LSUN-Bedrooms and FFHQ datasets, we use 200 denoising time steps. For class-conditional image generation on the ImageNet dataset, we employ the default DDIM sampler with 20 time steps and a guidance scale of 3.0. All experiments are implemented using PyTorch and conducted on a single NVIDIA A100 GPU.

**Evaluation metrics.** We evaluate the performance of diffusion models using Fréchet Inception Distance (FID) (Heusel et al., 2017) and sFID (Salimans et al., 2016) across all experiments for a fair comparison with previous works (Shang et al., 2023; Li et al., 2023; Huang et al., 2024). FID quantifies the difference between the Inception image features of synthetic and real images. On the other hand, sFID uses mid-level Inception features to better capture the spatial distribution similarity. For consistency, we compute the metrics using $50,000$ generated samples, in line with the settings in previous works (Shang et al., 2023; Li et al., 2023; Huang et al., 2024).

### 4.2 COMPARISON WITH THE STATE-OF-THE-ART METHODS

We compare our proposed method with the state-of-the-art approaches for PTQ on diffusion models, including PTQ4DM (Shang et al., 2023), Q-Diffusion (Li et al., 2023), PTQD (He et al., 2023), TFMQ-DM (Huang et al., 2024), APQ-DM (Wang et al., 2024), and TCAQ-DM (Huang et al., 2025). The results of competitors are taken from TFMQ-DM (Huang et al., 2024) and TCAQ-DM (Huang et al., 2025); the APQ-DM (Wang et al., 2024) results are reproduced from the official implementation. We conduct experiments on the CIFAR-10 $32 \times 32$, LSUN-Bedrooms $256 \times 256$, and FFHQ $256 \times 256$ datasets for unconditional image generation, and on ImageNet $256 \times 256$ dataset for class-conditional image generation, following the same experimental settings as (Huang et al., 2024).

**Unconditional image generation.** We conduct experiments including DDPM on the CIFAR-10 $32 \times 32$ dataset and LDM-4 on LSUN-Bedrooms $256 \times 256$ and FFHQ $256 \times 256$ datasets, using the DDIM sampler (Song et al., 2021a) with 100, 200 and 200 time steps, respectively. As shown in Table 1 and Table 2, our method achieves the best performance on CIFAR-10 $32 \times 32$ and LSUN-Bedrooms $256 \times 256$ datasets across most bit-width settings. The improvement is most evident in low bit-width settings. Specifically, on the CIFAR-10 $32 \times 32$ dataset, our method achieves an FID score improvement of 0.52 and 0.53 over the TFMQ-DM in the W4A32 and W4A8 settings, respectively. On the LSUN-Bedrooms $256 \times 256$ dataset, our proposed method achieves FID improvements over TFMQ-DM by 0.44 and 0.40 in the W4A32 and W4A8 settings, respectively. Meanwhile, on the FFHQ $256 \times 256$ dataset, our proposed method significantly reduces the FID score over TFMQ-DM by 0.69 and 0.63 in the W4A32 and W4A8 settings, respectively.

---

[1]https://github.com/facebookresearch/higher
[2]https://github.com/toshas/torch-fidelity

Table 2: Quantization results for unconditional and class-conditional image generation with LDM-4 on LSUN-Bedrooms $256 \times 256$, FFHQ $256 \times 256$, and ImageNet $256 \times 256$.

| Methods | Bits (W/A) | LSUN-Bedrooms | | FFHQ | | ImageNet | |
|---|---|---|---|---|---|---|---|
| | | FID↓ | sFID↓ | FID↓ | sFID↓ | FID↓ | sFID↓ |
| Full Prec. | 32/32 | 2.98 | 7.09 | 9.36 | 8.67 | 10.91 | 7.67 |
| PTQ4DM (Shang et al., 2023) | | 4.83 | 7.94 | 11.74 | 12.18 | - | - |
| Q-Diffusion (Li et al., 2023) | | 4.20 | 7.66 | 11.60 | 10.30 | 11.87 | 8.76 |
| PTQD (He et al., 2023) | 4/32 | 4.42 | 7.88 | 12.01 | 11.12 | 11.65 | 9.06 |
| TFMQ-DM (Huang et al., 2024) | | 3.60 | 7.61 | 9.89 | **9.06** | 10.50 | 7.98 |
| TCAQ-DM (Huang et al., 2025) | | 3.55 | 7.54 | - | - | 10.5 | **6.66** |
| Ours | | **3.16** | **6.92** | **9.20** | 9.69 | **10.10** | 7.32 |
| PTQ4DM (Shang et al., 2023) | | 4.75 | 9.59 | 10.73 | 11.65 | - | - |
| Q-Diffusion (Li et al., 2023) | | 4.51 | 8.17 | 10.87 | 10.01 | 12.80 | 9.87 |
| PTQD (He et al., 2023) | 8/8 | 3.75 | 9.89 | 10.69 | 10.97 | 11.94 | 8.03 |
| TFMQ-DM (Huang et al., 2024) | | 3.14 | 7.26 | 9.46 | **8.73** | 10.79 | 7.65 |
| TCAQ-DM (Huang et al., 2025) | | 3.11 | 7.34 | - | - | 10.58 | 7.54 |
| Ours | | **3.08** | **7.18** | **9.16** | 9.59 | **10.75** | **7.63** |
| PTQ4DM (Shang et al., 2023) | | 20.72 | 54.30 | 11.83 | 12.91 | - | - |
| Q-Diffusion (Li et al., 2023) | | 6.40 | 17.93 | 11.45 | 11.15 | 10.68 | 14.85 |
| PTQD (He et al., 2023) | 4/8 | 5.94 | 15.16 | 11.42 | 11.43 | 10.40 | 12.63 |
| TFMQ-DM (Huang et al., 2024) | | 3.68 | 7.65 | 9.97 | **9.14** | 10.29 | 7.35 |
| TCAQ-DM (Huang et al., 2025) | | 3.65 | 7.64 | - | - | 9.97 | 7.67 |
| Ours | | **3.28** | **7.05** | **9.34** | 9.74 | **10.01** | **7.21** |

**Class-conditional image generation.** For the ImageNet $256 \times 256$ experiments, we use LDM-4 with the DDIM sampler (Song et al., 2021a) (20 steps) to assess the performance of the quantized model. The results of the competitors are taken from TFMQ-DM (Huang et al., 2024). As shown in Table 2, our method outperforms the compared methods across most settings. Specifically, the proposed method achieves significant improvements over TFMQ-DM (Huang et al., 2024) in the W4A32 setting, with gains of $0.4$ and $0.66$ in FID and sFID, respectively.

**Visualization of the learned $\lambda$ and $\omega$.** Figure 3a and Figure 3b show the visualization of the learned frequency weight $\lambda$ and sample weight $\omega$, respectively. As shown, the weight of low-frequency component ($\lambda_0$) decreases, while the weights of high-frequency components ($\lambda_1, \lambda_2, \lambda_3$) increase as the timestep decreases. For the learned sample weight $\omega$, as time steps decrease, the normalized weights of samples become more variable and tend to increase, which indicates that the images generated at later time steps are often more important than those generated at earlier time steps for quantized diffusion models.

Table 3: The effects of sample and frequency weighting, and the regularization term $\mathcal{L}_{\text{Reg}}$ on LSUN-Bedrooms $256 \times 256$.

| Methods | Bits (W/A) | LSUN-Bedrooms $256 \times 256$ | |
|---|---|---|---|
| | | FID↓ | sFID↓ |
| Full Prec. | 32/32 | 2.98 | 7.09 |
| TFMQ-DM (Huang et al., 2024) (Baseline) | | 3.68 | 7.65 |
| TFMQ-DM + Sample weighting | | 3.47 | 7.20 |
| TFMQ-DM + Frequency weighting | 4/8 | 3.38 | 7.39 |
| Ours (without $\mathcal{L}_{\text{Reg}}$) | | 3.41 | 7.18 |
| Ours (sample and frequency weighting) | | **3.28** | **7.05** |

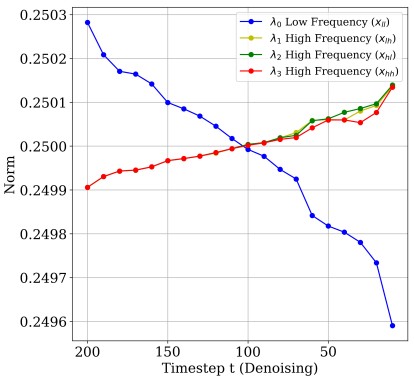
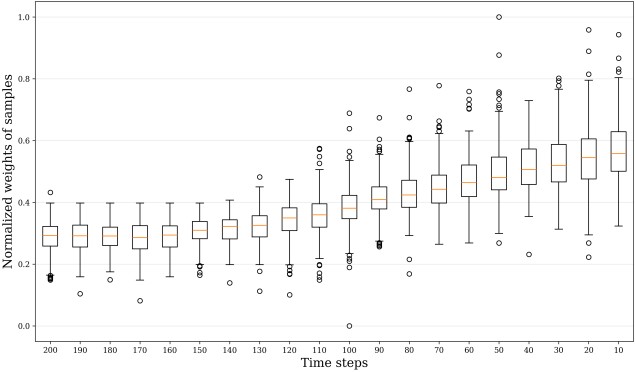

(a) Visualization of the learned frequency weight vector $\lambda$ for different frequency components over denoising timesteps.

(b) Box plot of the learned sample weights $\omega$ across time steps. The orange line indicates the median weights of the samples at each time step, while the circles represent outlier values outside the typical value range.

Figure 3: Visualization of learned weights.

## 4.3 ABLATION STUDIES

In this section, we conduct ablation studies to analyze the impact of each proposed component in our framework and the effects of the regularization terms. The ablation studies for the hyper-parameters $\beta$ in Eq. (7) and $\gamma$ in Eq. (12) are provided in the supplementary materials due to space constraints.

**The effects of the proposed sample and frequency meta weighting.** To evaluate the effectiveness of each component, we conduct an ablation study on the LSUN-Bedrooms $256 \times 256$ dataset with the W4A8 setting, using the LDM-4 model and a DDIM sampler. Table 3 shows that using either the sample weighting or frequency weighting component alone improves the performance of the quantized model. Additionally, combining these two strategies results in additional performance improvements. These results indicate the effectiveness of our proposed approach.

**The effects of the regularization term** $\mathcal{L}_{\text{Reg}}$ **in Eq.** (7). To validate the impact of the regularization term $\mathcal{L}_{\text{Reg}}$ on the quantized model performance, we conduct ablation studies on the LSUN-Bedrooms $256 \times 256$ dataset using the W4A8 quantization setting. The experiments are performed on the LDM-4 model with a DDIM sampler, with and without $\mathcal{L}_{\text{Reg}}$. As shown in Table 3, incorporating the regularization term $\mathcal{L}_{\text{Reg}}$ in Eq. (7) improves both the FID and sFID scores, showing the effectiveness of the proposed regularization loss function.

## 5 CONCLUSION

In this work, we present a novel approach for post-training quantization of diffusion models that incorporates sample and frequency weighting. Our method simultaneously optimizes the contributions of calibration samples and the weighting of frequency components at each time step to effectively quantize the noise estimation networks. By automatically learning optimal weights for calibration samples, our approach prioritizes important samples and enhances the performance of the quantized model. Additionally, by learning to weight frequency components in the frequency loss for each time step, we encourage the quantized models to better mimic the frequency components of data generated from their full-precision counterparts. Extensive experimental results show that our proposed method consistently outperforms the state-of-the-art PTQ approaches for diffusion models, demonstrating its effectiveness across different datasets.

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

**The statement on the use of large language models.** Large Language Models (LLMs) were used solely for grammar correction and language polishing of this manuscript. All research ideas, experimental design and data analysis were conducted entirely by the authors, and the use of LLMs does not impact the reproducibility or validity of our findings.

# A APPENDIX

## A.1 MORE EXPERIMENTS

Table A.1: Quantization results for unconditional image generation with DDIM on LSUN-Bedrooms $256 \times 256$. The notation $*$ indicates that we use the small real validation set.

| Methods | Bits (W/A) | LSUN-Bedrooms $256 \times 256$ | |
| --- | --- | --- | --- |
| | | FID↓ | sFID↓ |
| Full Prec. | 32/32 | 2.98 | 7.09 |
| PTQ4DM (Shang et al., 2023) | | 4.83 | 7.94 |
| Q-Diffusion (Li et al., 2023) | 4/32 | 4.20 | 7.66 |
| TFMQ-DM (Huang et al., 2024) | | 3.60 | 7.61 |
| Ours | | 3.16 | **6.92** |
| Ours* | | **3.12** | 7.15 |
| PTQ4DM (Shang et al., 2023) | | 20.72 | 54.30 |
| Q-Diffusion (Li et al., 2023) | 4/8 | 6.40 | 17.93 |
| TFMQ-DM (Huang et al., 2024) | | 3.68 | 7.65 |
| Ours | | 3.28 | **7.05** |
| Ours* | | **3.26** | 7.19 |

**Experiments with small real validation set.** We conduct additional experiments with the small real validation set from the LSUN-Bedrooms dataset for computing $\mathcal{L}_{val}$ in Eq. (7). Specifically, instead of using the subset of generated images from the full-precision model, we randomly sample 32 images from the LSUN-Bedrooms dataset. After that, we create corresponding images at different time steps by adding Gaussian noise to the sampled images to form a validation set. Following the forward diffusion process (Ho et al., 2020), the noisy image $\mathbf{x}_t$ at time step $t$ is defined as:

$$\mathbf{x}_t = \sqrt{\bar{\alpha}_t}\mathbf{x}_0 + \sqrt{1 - \bar{\alpha}_t}\boldsymbol{\epsilon}, \tag{13}$$

where $\mathbf{x}_0$ is the original clean image, $\bar{\alpha}_t = \prod_{i=1}^{t} \alpha_i$ represents the cumulative product of noise schedule coefficients, and $\boldsymbol{\epsilon} \sim \mathcal{N}(0, \mathbf{I})$ is random Gaussian noise.

The results are shown in Table A.1. As shown, using either the generated images or the real images, our proposed method consistently outperforms TFMQ-DM (Huang et al., 2024). This may be because the full-precision model is trained on real images, so using either real images or generated images for validation yields similar performance outcomes.

**Experiments with an alternative method for frequency loss in Eq.** (6)**.** We explore an alternative approach that computes the frequency loss based on the approximated final samples ($t = 0$), which can be estimated directly from any generated sample $x_i$ at time step $t_i$. Following DDPM (Ho et al., 2020), given an intermediate generated sample $\boldsymbol{x}_i$ at the time step $t_i$, the corresponding generated sample $\widehat{\boldsymbol{x}}$ at the time step $t = 0$ can be estimated as follows:

$$\widehat{\boldsymbol{x}} = (\boldsymbol{x}_i - \sqrt{1 - \bar{\alpha}_{t_i}}\boldsymbol{\epsilon}_\theta(\boldsymbol{x}_i, t_i))/\sqrt{\bar{\alpha}_{t_i}}, \tag{14}$$

where $\bar{\alpha}_{t_i} = \prod_{i=1}^{t_i} \alpha_i$, and $\alpha_1, \ldots, \alpha_T$ are predefined variance schedules. Using the above approximation, in each iteration, we acquire approximations for the final generated images of the full-precision model and the quantized model $\widehat{\boldsymbol{x}}_Q$ and $\widehat{\boldsymbol{x}}_{FP}$. The frequency loss $\mathcal{L}_F(.)$ is defined as:

$$\mathcal{L}_F(\theta_Q, x_i, \lambda) = \mathcal{L}_f\left(\widehat{\boldsymbol{x}}_Q, \widehat{\boldsymbol{x}}_{FP}, \lambda_{t_i}\right). \tag{15}$$

Table A.2: Quantization results for unconditional image generation with DDIM on CIFAR-10 $32 \times 32$. The notation $*$ indicates that we use the alternative frequency loss in Eq. (15).

| Methods | Bits (W/A) | CIFAR-10 $32 \times 32$ | |
| --- | --- | --- | --- |
| | | FID↓ | sFID↓ |
| Full Prec. | 32/32 | 4.23 | 4.41 |
| PTQ4DM (Shang et al., 2023) | | 5.65 | - |
| Q-Diffusion (Li et al., 2023) | | 5.08 | 4.98 |
| TFMQ-DM (Huang et al., 2024) | 4/32 | 4.73 | - |
| Ours | | **4.21** | 4.47 |
| Ours* | | 4.29 | **4.45** |
| PTQ4DM (Shang et al., 2023) | | 5.69 | - |
| Q-Diffusion (Li et al., 2023) | | 4.78 | 4.75 |
| TFMQ-DM (Huang et al., 2024) | 8/8 | 4.24 | - |
| Ours | | 4.15 | 4.36 |
| Ours* | | **4.09** | **4.34** |
| PTQ4DM (Shang et al., 2023) | | 5.14 | - |
| Q-Diffusion (Li et al., 2023) | | 4.98 | 5.68 |
| TFMQ-DM (Huang et al., 2024) | 4/8 | 4.78 | - |
| Ours | | **4.25** | **4.46** |
| Ours* | | 4.31 | 4.57 |

We evaluate the alternative frequency loss on the CIFAR-10 $32 \times 32$ dataset. As shown in Table A.2, this alternative approach yields comparable results. However, from our experiments we observe that it requires up to three times the computational cost compared to the original frequency loss in Eq. (6). Therefore, we use the frequency loss defined in Eq. (6) for the results in the main paper and the remaining sections in the supplementary materials.

## A.2 HYPER-PARAMETER SETTINGS

Regarding the hyper-parameters $\beta$ in Eq. (7) and $\gamma$ in Eq. (12) in the main paper, $\beta$ is applied to the regularization loss $\mathcal{L}_{\text{Reg}}$, while $\gamma$ controls the contribution of the frequency loss to the final objective for quantizing diffusion models.

Table A.3: Ablation studies for the hyper-parameter $\gamma$ of the frequency loss in Eq. (12). The results are on the CIFAR-10 dataset with the W4A32 setting.

| $\gamma$ | 0.05 | 0.1 | 0.2 | 0.3 | 0.5 | 0.8 | 1.0 |
| --- | --- | --- | --- | --- | --- | --- | --- |
| FID↓ | 4.41 | **4.21** | 4.29 | 4.35 | 4.58 | 4.76 | 4.71 |
| sFID↓ | 4.50 | **4.47** | 4.53 | 4.56 | 4.67 | 4.98 | 4.91 |

**Ablation studies for the hyper-parameter $\gamma$ in Eq. (12).** We vary the value of $\gamma$ from 0.05 to 1 and fix the value of $\beta = 0.05$, and evaluate the performance of the model on the CIFAR-10 dataset with the W4A32 setting. The results are shown in Table A.3.

As shown in the table, the performance is stable across different choices of $\gamma$. The performance is slightly better with $\gamma = 0.1$, whereas larger $\gamma$ values (e.g., $\gamma = 1$) may slightly degrade performance. This indicates that the proposed method is not sensitive to the choice of $\gamma$ and $\beta$.

**Ablation studies for the hyper-parameter $\beta$ in Eq. (7).** We vary the value of $\beta$ from 0.01 to 0.1 and fix the value of $\gamma = 0.1$. The experiments are also conducted on the CIFAR-10 dataset with the W4A32 setting. The results are shown in Table A.4. As shown in the table, the performance is stable across different choices of $\beta$. The performance is slightly better with $\beta = 0.05$. This indicates that the proposed method is not sensitive to the choice of $\beta$.

Table A.4: Ablation studies for the hyper-parameter $\beta$ of the $\mathcal{L}_{\text{Reg}}$ in Eq. (7). The results are on the CIFAR-10 dataset with W4A32 setting.

| $\beta$ | 0.01 | 0.02 | 0.03 | 0.05 | 0.08 | 0.1 |
|---|---|---|---|---|---|---|
| FID↓ | 4.25 | 4.34 | 4.31 | **4.21** | 4.68 | 4.73 |
| sFID↓ | 4.6 | 4.56 | 4.52 | **4.47** | 4.55 | 4.58 |

Table A.5: Quantization results for unconditional image generation with LDM-4 on LSUN-Bedrooms $256 \times 256$.

| Methods | Bits (W/A) | LSUN-Bedrooms $256\times256$ | |
|---|---|---|---|
| | | FID↓ | sFID↓ |
| Full Prec. | 32/32 | 2.98 | 7.09 |
| TFMQ-DM (Huang et al., 2024) | | 3.68 | 7.65 |
| Ours (FFT) | 4/8 | 3.45 | 7.20 |
| Ours (DWT) | | **3.28** | **7.05** |

## A.3 COMPARE WITH OTHER FREQUENCY TRANSFORMATION METHODS

DWT is a widely used frequency analysis method. It effectively separates and analyzes low and high frequencies from other frequency transforms, such as Fast Fourier Transform (FFT). We conduct additional experiments using FFT and leveraging Focal Frequency Loss [3] for $\mathcal{L}_F$ in Eq. (12). The results in Table A.5 demonstrate that leveraging the frequency domain with either DWT or FFT for the PTQ of diffusion models outperforms the baseline TFMQ-DM (Huang et al., 2024), with DWT showing superior results.

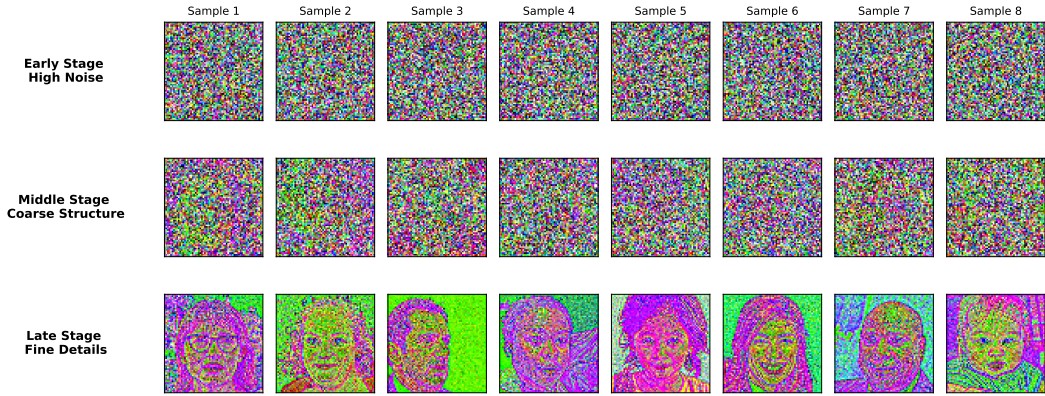

Figure A.1: Visualization of image evolution of calibration dataset sampled from the FFHQ $256 \times 256$ dataset.

## A.4 VISUALIZATION OF CALIBRATION DATASET

We visualize the calibration dataset sampled from the full-precision diffusion model trained on the FFHQ $256 \times 256$ dataset. As shown in Figure A.1, eight different samples are displayed at three representative denoising stages: early stage (high noise), middle stage (coarse structure formation), and late stage (fine detail refinement).

These visualizations reveal important frequency characteristics across the denoising process. At early timesteps, the images predominantly contain noise with minimal structural information, indicating that the diffusion model primarily works with noisy, unstructured patterns. During the middle stage, the model recovers coarse structures and overall composition. At late timesteps, the model refines details such as textures, edges, and fine-grained features.

---

[3]https://github.com/EndlessSora/focal-frequency-loss

## A.5 VISUALIZATION OF GENERATED IMAGES

We visualize sample images generated from the full-precision model, as well as from quantized models obtained using the Q-Diffusion (Li et al., 2023) method, the TFMQ (Huang et al., 2024) method, and our proposed method with the W4A8 setting, all initialized with a fixed random seed. As shown in Figure A.2 and Figure A.3, our proposed method generates images that closely match those of the full-precision models, demonstrating the effectiveness of our approach.

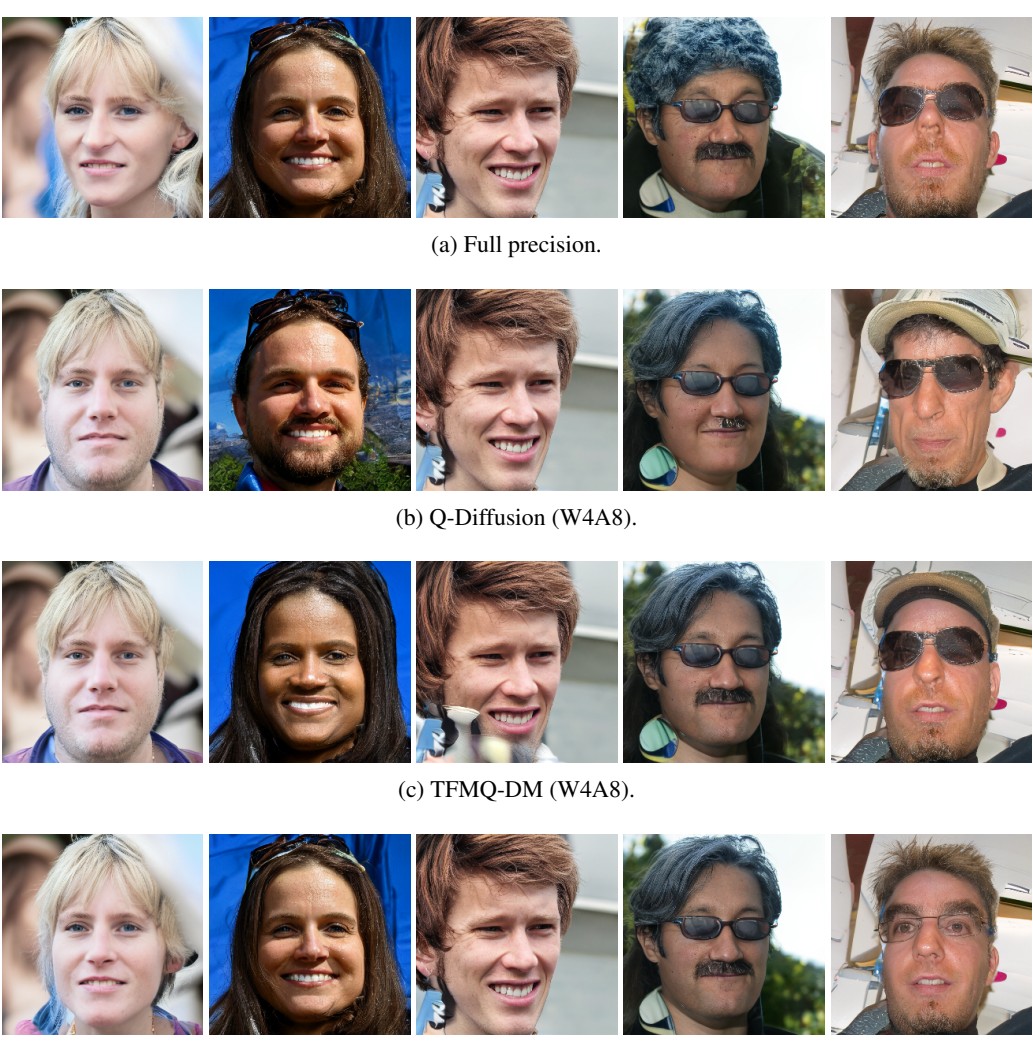

(a) Full precision.

(b) Q-Diffusion (W4A8).

(c) TFMQ-DM (W4A8).

(d) Our proposed method (W4A8).

Figure A.2: Generated samples from (a) full-precision LDM-4, (b) Q-Diffusion (W4A8), (c) TFMQ-DM (W4A8), and (d) our proposed method (W4A8) on FFHQ $256 \times 256$ dataset with a fixed random seed.

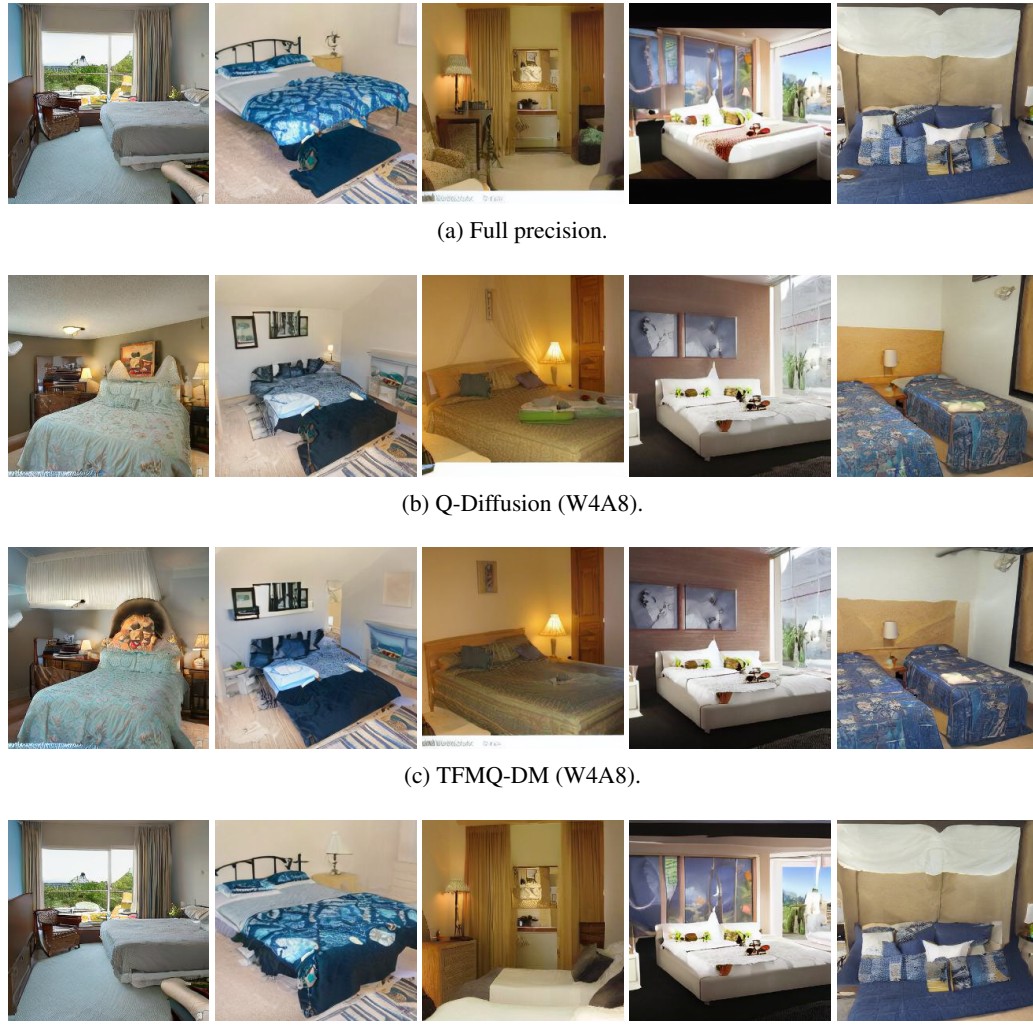

(a) Full precision.

(b) Q-Diffusion (W4A8).

(c) TFMQ-DM (W4A8).

(d) Our proposed method (W4A8).

Figure A.3: Generated samples from (a) full-precision LDM-4, (b) Q-Diffusion (W4A8), (c) TFMQ-DM (W4A8), and (d) our proposed method (W4A8) on LSUN-Bedrooms $256 \times 256$ dataset with a fixed random seed.

