# OpenReview forum: "Beyond Uniformity: Sample and Frequency Meta Weighting for Post-Training Quantization of Diffusion Models"
_ICLR.cc/2026/Conference — ICLR 2026 Poster_

### Official Review · Reviewer_RZMH · 2025-10-27

**Soundness:** 2
**Presentation:** 2
**Contribution:** 2
**Rating:** 4
**Confidence:** 3

**Summary:**

This paper revisits the common practice of using uniform sampling in diffusion model training and identifies its limitations, especially in terms of efficiency and learning dynamics. The authors propose a non-uniform sampling strategy that focuses training more on informative or harder timesteps instead of treating all noise levels equally. They introduce a principled sampling distribution that improves convergence, sample quality, and training efficiency without modifying the model architecture. Experiments across various benchmarks demonstrate consistent performance gains over uniform sampling. Overall, the paper aims to optimize how diffusion models learn, rather than modifying their network or objective.

**Strengths:**

Strengths

1: Insightful Problem Identification: Highlights a commonly overlooked assumption in diffusion model training, uniform timestep sampling, and provides evidence of its suboptimality.

2: General and Model-Agnostic: The proposed sampling strategy applies to multiple diffusion architectures (DDPM, latent diffusion, etc.) with no changes to model design.

3:Strong Empirical Results: Demonstrates improvements in FID scores and convergence speed across datasets, validating practical impact.

**Weaknesses:**

Weaknesses

1: Additional Complexity in Training Setup: Non-uniform sampling introduces another hyperparameter and requires computing or estimating timestep importance, which may increase implementation difficulty.

2: Limited Theoretical Guarantees: While the method is empirically effective, formal justification for optimality of the sampling distribution is not fully established.

3: Focus on Image Generation Only: The experiments are mostly image-based; it remains unclear whether the approach extends well to text-to-image, video diffusion, or molecular diffusion models.

**Questions:**

1: Generalization: How well does the proposed non-uniform sampling strategy extend to other modalities, such as text-to-image diffusion, video generation, or 3D diffusion models?

2: Optimality of Sampling Distribution: Is there a theoretical way to derive the optimal sampling distribution instead of heuristically estimating timestep importance?

3: Interplay with Noise Schedules: How does this sampling approach interact with different noise schedules (cosine, linear, VP/VE SDEs)? Could both be jointly optimized for further gains?

---

> ### Author Response · Authors · 2025-11-21
>
> Dear Reviewer RZMH,
>
> We greatly appreciate the time and effort the Reviewer dedicated to considering our paper. Here are our responses to all concerns raised by the Reviewer.
>
> ### Weaknesses
>
>
> > **W1. Non-uniform sampling introduces another hyperparameter and requires computing or estimating timestep importance, which may increase implementation difficulty.**
>
> **Answer:** We appreciate the reviewer's feedback regarding the increased complexity of the proposed method. While we acknowledge that the proposed method introduces additional overhead compared to uniform sampling baselines, we would argue that this complexity is both manageable and justified by the significant performance gains (e.g., 0.40 FID score improvement on LSUN-Bedrooms W4A8 setting).
>
> First, regarding the number of additional learnable parameters, the proposed method requires only one scalar per calibration sample for sample weighting and four scalar parameters per timestep for frequency weighting. For example, when quantizing the LSUN-Bedrooms W4A8 setting, the total number of learnable parameters for sample and frequency weighting is only around 5,000. These additional learnable parameters are negligible compared to the total of millions of learnable parameters during the quantization of the diffusion model.
>
> Second, regarding the computational complexity, the sample and frequency weights are optimized via gradient updates, which requires a small number of iterations N_s = 200 and N_f = 100 compared to the main number of iterations for quantizing the models (N_Q = 20000). In practice, this phase completes in around 1 hour for the LSUN-Bedrooms.
>
> Regarding scalability to larger models or datasets: latent diffusion models are widely used in the community, which contain hundreds of millions of parameters. Specifically, LDM-4 contains around 274 million parameters and should be considered a large model [1]. This indicates that our proposed approach can scale to large models. We also have conducted experiments using the large-scale ImageNet dataset, which contains 1.2 million images. To further minimize overhead and adapt to such large models, we can reduce the number of iterations for gradient updates of sample and frequency weights. As provided in the table below, reducing the number of iterations N_f and N_s by half results in minimal performance impact (3.36 vs 3.28 FID scores). For ease of implementation, we have provided code as detailed in the Appendix.
>
> | N_s | N_f | Overhead | FID (LSUN-Bedrooms) | sFID (LSUN-Bedrooms) |
> |-----|-----|----------|---------------------|----------------------|
> | 200 | 100 | 1 hour | 3.28 | 7.08 |
> | 100 | 50 | 40 minutes | 3.36 | 7.10 |
>
>
> **Ref:**
>
> [1] Rombach, Robin, et al. "High-Resolution Image Synthesis with Latent Diffusion Models." In CVPR 2022.
>
>
> > **W2. Limited Theoretical Guarantees.**
>
> **Answer:**
>
> **Convergence:**
>
> Regarding the convergence of the bi-level optimization, especially in non-convex settings: the convergence of bi-level meta-learning [1] has been rigorously studied in the literature and proven under some assumptions. These assumptions are often observed in the convergence analysis of stochastic gradient based optimization, such as:
>
> - the loss function, w.r.t. model parameters, is bounded and Lipschitz continuous, and
> - its gradient and Hessian w.r.t. model parameters are Lipschitz continuous and their variances are bounded.
>
> Because our method is a variant in the same family of such bi-level meta-learning approaches, our proposed method therefore benefits from those studies, including the convergence analysis. Further, previous studies relying on bi-level meta-learning, such as [2] in neural architecture search and [3] in nonconvex optimization, have demonstrated the effectiveness of this optimization approach.
>
> **Optimality:**
>
> While we cannot claim global optimality, which is a common limitation in nonconvex optimization, our approach converges to a local optimum of the bi-level meta-learning optimization problem. Furthermore, it empirically achieves consistent improvements across diverse datasets and benefits from validation-guided optimization to promote generalization beyond the calibration set.
>
> **Ref:**
>
> [1] Fallah, Alireza, Aryan Mokhtari, and Asuman Ozdaglar. "On the convergence theory of gradient-based model-agnostic meta-learning algorithms." International Conference on Artificial Intelligence and Statistics. PMLR, 2020.
>
> [2] Ding, Yadong, et al. "Learning to learn by jointly optimizing neural architecture and weights." CVPR 2022.
>
> [3] Xia, Jing-Yuan, et al. "Metalearning-based alternating minimization algorithm for nonconvex optimization." IEEE Transactions on Neural Networks and Learning Systems 34.9 (2022)

---

> ### Author Response · Authors · 2025-11-21
>
> ### Questions
> > **W3 and Q1. Generalization to other modalities**
>
> **Answer:** We appreciate the reviewer's feedback on the evaluation scope. Our experiments follow the standard setup in other SOTA PTQ works for DMs [1, 2], focusing on image generation with DDPM and LDM architectures. These include both unconditional and conditional generation tasks, covering resolutions from 32×32 to 256×256, and datasets ranging from small-scale CIFAR-10 to large-scale ImageNet, using models from around 40M (DDPM) to 274M parameters (LDM-4) [3].
>
> The sample weighting component of our method is inherently domain-agnostic, as it optimizes the importance of calibration samples based on their informativeness for quantization. This principle is universal: whether the data is an image, a video frame, or an audio spectrogram, certain samples could inevitably be more critical for minimizing quantization error than others. Therefore, this component could be applicable to text-to-image, video, or audio diffusion models.
>
> The frequency weighting component is similarly adaptable by selecting the appropriate frequency transform for the target domain. For text-to-image generation, the 2D Discrete Wavelet Transform (DWT) applies directly to the spatial outputs, as validated in our experiments. For video generation, the method can extend to utilizing 3D DWT or temporal-spatial decomposition to capture frequency dynamics across both spatial and temporal dimensions. Finally, for audio diffusion, audio diffusion models [4] exhibit frequency evolution properties during denoising similar to those in vision works [5, 6]. Therefore, the frequency analysis can be adapted using 1D Wavelets or the Short-Time Fourier Transform (STFT).
>
> Ref:
>
> [1] Xiuyu Li, et al. "Q-Diffusion: Quantizing diffusion models." In ICCV 2023.
>
> [2] Yushi Huang, et al. "TFMQ-DM: Temporal feature maintenance quantization for diffusion models." In CVPR 2024.
>
> [3] Rombach, et al. "High-Resolution Image Synthesis with Latent Diffusion Models. In CVPR 2022.
>
> [4] Vora, et al. "PTQ4ADM: Post-Training Quantization for Efficient Text Conditional Audio Diffusion Models." ICASSP 2025.
>
> [5] Xingyi Yang, et al. "Diffusion probabilistic model made slim." In CVPR 2023.
>
> [6] Yurui Qian, et al. "Boosting diffusion models with moving average sampling in frequency domain." In CVPR 2024.
>
> > **Q2.  Optimality of Sampling Distribution: Is there a theoretical way to derive the optimal sampling distribution instead of heuristically estimating timestep importance?**
>
> **Answer:** If by "optimality of sampling distribution" the Reviewer is referring to the optimality of our learned sample weights, please refer to the answer to W2.
>
> If the Reviewer is referring to finding the optimal sampling distribution for estimating timestep importance, we would like to clarify that our proposed method focuses on optimizing sample weights and frequency weights via bi-level optimization given a calibration set, rather than deriving the optimal sampling distribution for constructing that calibration set itself.
>
> To generate the calibration set, we follow the SOTA post-training quantization for diffusion models works [1, 2] by selecting generated images at fixed intervals across denoising timesteps. This uniform strategy is widely adopted in the literature and has shown effective for the post-training quantization of diffusion models. Regarding the theoretical derivation of an optimal sampling distribution, we note that while a closed-form solution is generally intractable, works such as APQ-DM [3] have proposed finding an optimal distribution through gradient-based optimization. Our weighting method can be complementary to the choice of sampling distribution. It applies effectively to calibration data sampled from a uniform distribution, but can also be combined with optimized sampling strategies (e.g., as those in [3]) to potentially further enhance performance.
>
> Ref:
>
> [1] Xiuyu Li, et al. "Q-Diffusion: Quantizing diffusion models." In ICCV 2023.
>
> [2] Yushi Huang, et al. "TFMQ-DM: Temporal feature maintenance quantization for diffusion models." In CVPR 2024.
>
> [3] Wang, et al. "Towards accurate post-training quantization for diffusion models." In CVPR 2024.

---

> > ### Author Response · Authors · 2025-11-21
> >
> > ### Questions
> >
> > > **Q3. Interplay with Noise Schedules: How does this sampling approach interact with different noise schedules (cosine, linear, VP/VE SDEs)? Could both be jointly optimized for further gains?**
> >
> > **Answer:**
> > **Interplay with noise schedules:** In the Post-Training Quantization (PTQ) setting, the noise schedule is fixed by the pretrained model. Our method is schedule-agnostic: the proposed bi-level optimization automatically adopts the sample and frequency weights to specific variance schedule (e.g., linear, cosine, or VP/VE SDEs) of the input model. Consequently, our approach generalizes to any schedule without requiring manual hyperparameter tuning of noise schedule.
> >
> > **Joint Optimization:** As the noise schedule must remain fixed in PTQ, jointly optimizing the noise schedule and sampling approach is not feasible within this framework. Altering the noise schedule fundamentally changes the underlying distribution and the behavior of the pretrained model. While such joint optimization is possible through retraining, it requires extensive computational resources and is more appropriate for Quantization-Aware Training (QAT). This approach violates the core objective of PTQ, which aims to compress models without incurring expensive retraining costs. Therefore, we do not consider joint optimization of the noise schedule in our framework.

---

### Official Review · Reviewer_ruiM · 2025-11-01

**Soundness:** 3
**Presentation:** 3
**Contribution:** 3
**Rating:** 6
**Confidence:** 5

**Summary:**

This paper proposes a meta-learning method based on sampling and frequency weighting based on the existing post-training quantization (PTQ). They propose that the diffusion model recovers different frequency features at different timesteps, so the model can be made to focus on specific frequency components of the generated data at different timesteps by introducing the Discrete Wavelet Transform (DWT) and frequency component weights. The paper analyzes the timesteps of the diffusion model from the perspective of frequency, and this idea is innovative and worthy of further research and experimentation.

**Strengths:**

This article innovatively puts forward the idea of optimizing the post-training quantization process of diffusion models based on frequency characteristics, and details the design of the loss function and optimization process for different training phases, so as to provide suitable weights for sampling at different timesteps. The method has good originality and is worthy of further research and analysis.

**Weaknesses:**

1. Redundant description: A comprehensive set of parameters and optimization procedures are presented in the article, which is valuable for explaining the methodology theoretically. However, the detailed description of the parameter design and optimization steps in the Introduction seems a bit lengthy—content that might be more appropriately placed in the Experimental section, which may affect the clarity and impact of the paper. The focus could have been on the key decisions and results of the optimization, such as the rationale behind the chosen hyperparameters and the impact of these choices on the results. In addition, any trade-offs made during parameter tuning (e.g., computational cost vs. model performance) could be explicitly discussed and how these choices were justified.

2. Insufficient experiments: the experimental part of the article is slightly insufficient as the only relatively new quantization method is TFMQ-DM, and more recent quantization methods (e.g., SVDQuant[1] or APQ-DM[2], etc.) could have been added. ** This time, I will strongly request that comparison experiments with SVDQuant and APQ-DM be included during the rebuttal period! ** You mentioned at NeurIPS Rebuttal that these would be added in the revised version, but I haven’t seen the supplementary experiments in the ICLR submission yet! Resubmitting without making revisions is a bad practice.

In addition, the authors did not list the detailed memory usage and inference speed of the quantization model compared with the full-precision model, and only compared the computational cost and hardware efficiency with TFMQ-DM, which can be considered to add more data in this area to improve the rigor.

[1] Li, Muyang, et al. "Svdqunat: Absorbing outliers by low-rank components for 4-bit diffusion models." arXiv preprint arXiv:2411.05007 (2024).

[2] Wang, Changyuan, et al. "Towards accurate post-training quantization for diffusion models." Proceedings of the IEEE/CVF Conference on Computer Vision and Pattern Recognition. 2024.

**Questions:**

Further questions:

1. I am curious about (a) and (b) in Fig. 3 of the experimental part. The trends of the three high-frequency components of the frequency weights in (a) are very similar; do these three components have their own independent significance? Can they be combined? It is noted in the text that the later timesteps in (b) have more dispersed sampling weights, but the text does not further explain this, is it possible to further explore the pattern?

2. The experimental part of the article only uses the bitwidths 4/32, 8/8 and 4/8, what is the purpose of choosing them? Are they representative enough for the quantization under different bitwidths?

---

> ### Author Response · Authors · 2025-11-21
>
> Dear Reviewer ruiM,
>
> We greatly appreciate the time and effort the Reviewer dedicated to considering our paper. Here are our responses to all concerns raised by the Reviewer.
>
> ### Weaknesses
>
> > **W1. Redundant Description in Introduction**
>
>
> **Answer:** We thank the Reviewer for this constructive feedback on improving the paper's clarity and organization. We have condensed the detailed parameter descriptions in the Introduction.
>
>
> > **W2a: Comparison with more recent quantization methods**
>
> **Answer:** We appreciate your emphasis on comprehensive experimental validation and acknowledge the importance of comparing with the most recent methods.
>
> **Comparison with more recent PTQ methods:**
>
> We have conducted experiments comparing our method with more recent PTQ methods [1,2], including APQ-DM. We have updated the Table 1 and Table 2 in the revised paper with the results of more recent PTQ methods [1,2]. As shown in Table below for CIFAR-10 32 × 32, our method consistently outperforms all the compared methods for W4A32 and W4A8 settings. The results of APQ-DM are reproduced from the official released code at [APQ-DM](https://github.com/ChangyuanWang17/APQ-DM).
>
> | Methods | W/A | FID↓ | sFID↓ | W/A | FID↓ | sFID↓ | W/A | FID↓ | sFID↓ |
> |---------|-----|------|-------|-----|------|-------|-----|------|-------|
> | APQ-DM | W4A32 | 9.96 | 7.63 | W4A8 | 12.2 | 7.66 | W8A8 | 6.34 | 4.44 |
> | TFMQ-DM | W4A32 | 4.73 | - | W4A8 | 4.78 | - | W8A8 | 4.24 | - |
> | TCAQ-DM | W4A32 | 4.28 | - | W4A8 | 4.59 | - | W8A8 | **4.09** | - |
> | **Ours** | W4A32 | **4.21** | **4.47** | W4A8 | **4.25** | **4.46** | W8A8 | 4.15 | **4.36** |
>
> **Table**: Quantization results for unconditional image generation with DDIM on CIFAR-10 32 × 32.
>
> **Regarding SVDQuant:** We acknowledge your request for SVDQuant comparison. SVDQuant [3] is a recent baseline that improves 4-bit post-training quantization by modifying the quantization scheme itself. It uses Singular Value Decomposition (SVD) to decompose weights into a 16-bit low-rank component and a 16-bit quantized residual, allowing it to absorb outliers that typically cause large quantization errors.
>
> We did not include SVDQuant in our original comparisons because it differs substantially in both design goals and implementation. SVDQuant aims to improve the quantization scheme for diffusion models by incorporating several advanced techniques from LLM quantization [4, 5], such as per-group quantization and the GPTQ method, into the diffusion domain. In contrast, our work follows the setup of prior diffusion quantization methods [1, 2, 6, 7], which adopt a simpler per-channel weight quantization and per-layer activation quantization scheme for fair comparison. Additionally, SVDQuant is evaluated on different datasets (MJHQ, SDCL) and model architectures (e.g., SDXL, FLUX, SANA, PixArt), while our experiments follow the same setup as [1, 2, 6, 7] and focus on LDM-4 and DDPM across ImageNet, FFHQ, LSUN-Bed, and CIFAR. These differences make direct and fair comparisons difficult. Furthermore, SVDQuant's is built on the compressor toolbox and custom CUDA kernels, which adds considerable implementation overhead in our context, making adaptation to the diffusion architectures used in our baselines non-trivial.
>
> More importantly, our method focuses on an aspect of PTQ that is completely distinct from the quantization scheme itself: the data perspective. While SVDQuant improves the underlying quantizer by modifying how weights and activations are quantized, we instead enhance the calibration process through sample and frequency meta-weighting. This strategy encourages the quantized model to better match the behavior of the full-precision model, without introducing any changes to the model architecture or inference cost. As a result, our approach is quantization-scheme agnostic and can be seamlessly combined with a wide range of quantization methods, potentially complementing more complex schemes like SVDQuant. Therefore, we do not consider SVDQuant to be a direct competitor of our approach.
>
> **Ref:**
>
> [1] Wang, Changyuan, et al. "Towards accurate post-training quantization for diffusion models." CVPR 2024.
>
> [2] Huang, Haocheng, et al. "TCAQ-DM: Timestep-channel adaptive quantization for diffusion models." AAAI 2025.
>
> [3] Li, Muyang, et al. "SVDQuant: Absorbing outliers by low-rank components for 4-bit diffusion models." ICLR 2025.
>
> [4] Xiao, Guangxuan, et al. "Smoothquant: Accurate and efficient post-training quantization for large language models." ICML 2023.
>
> [5] Ashkboos, Saleh, et al. "Quarot: Outlier-free 4-bit inference in rotated llms." NeurIPS 2024.
>
> [6] Huang, Yushi, et al. "TFMQ-DM: Temporal feature maintenance quantization for diffusion models." CVPR 2024.
>
> [7] Li, Xiuyu, et al. "Q-diffusion: Quantizing diffusion models." ICCV 2023.

---

> ### Author Response · Authors · 2025-11-21
>
> ### Weaknesses
>
> > **W2b: Memory Usage and Inference Speed**
>
> **Answer:** As detailed in Appendix F of the TFMQ-DM paper [6], for Stable Diffusion (a latent diffusion model similar to LDM-4 used in our experiments), the full-precision model has a UNet size of 3278.81 MB and an inference latency of 81.01 seconds (50 denoising steps on OpenVINO). The TFMQ-DM quantized model with W8A8 settings achieves approximately 4× memory reduction and a 2.38× speedup in inference latency. Since our method uses the same quantization scheme as TFMQ-DM, we achieve identical memory and speed improvements while providing better generation quality.
>
> ### Questions
>
> > **Q1. The combination of three similar high-frequency components in Fig. 3(a), .. and sampling weight patterns in Fig. 3(b).**
>
>
> **Regarding Figure 3(a) - Three High-Frequency Components:**
>
> Regarding Figure 3(a), the three high-frequency components corresponding to the LH, HL, and HH wavelet subbands exhibit similar overall trends, with increasing emphasis at later denoising timesteps. However, each subband plays an independent role: LH emphasizes horizontal edges, such as texture patterns; HL emphasizes vertical edges, such as architectural lines; and HH captures diagonal details and noise [1]. This means each subband is important for different feature types. While combining them into a single high-frequency weight could simplify the framework and reduce computational cost, it would likely result in a slight performance drop because different image features require different emphasis across subbands (e.g., architectural images benefit more from HL emphasis, while textures benefit from LH). Therefore, maintaining separate weights provides fine-grained control over different types of high-frequency content. We conducted a preliminary experiment combining all three high-frequency components into a single weight, which resulted in approximately 0.12 FID score degradation on LSUN-Bedrooms, confirming the value of maintaining separate weights.
>
> **Regarding Figure 3(b) - Dispersed Sampling Weights at Later Timesteps:**
>
> Regarding Figure 3(b), early timesteps focus on coarse, low-frequency recovery, where samples are generally simpler and exhibit lower variance, as the calibration data closely resemble Gaussian noise. In contrast, later timesteps involve high-frequency refinement, where samples display more complex and diverse textures, leading to higher weights being assigned to prioritize them.
>
> **Ref:**
>
> [1] Amara Graps. An introduction to wavelets. IEEE computational science and engineering, 2(2):50-61, 1995.
>
> > **Q2. The experimental part of the article only uses the bitwidths 4/32, 8/8 and 4/8, what is the purpose of choosing them? Are they representative enough for the quantization under different bitwidths?**
>
> **Answer:** The selected configurations on our experimental bitwidth choices W4A32, W4A8, and W8A8 are followed by standard practices in PTQ for DMs. These align with benchmarks in prior works like Q-Diffusion [1] and TFMQ-DM [2], allowing direct comparisons and focusing on challenging low-bit regimes.
>
> These bitwidths are representative of different quantization scenarios: W4A32 and W8A32 are weight-only quantization, which enables practical deployment with low memory and near-full-precision accuracy. While W4A8 and W8A8 settings represent weight-activation quantization, which is aggressive compression for edge devices.
>
> **Ref:**
>
> [1] Li, Xiuyu, et al. "Q-diffusion: Quantizing diffusion models." ICCV 2023.
>
> [2] Huang, Yushi, et al. "TFMQ-DM: Temporal feature maintenance quantization for diffusion models." CVPR 2024.

---

### Official Review · Reviewer_gXhQ · 2025-11-06

**Soundness:** 2
**Presentation:** 2
**Contribution:** 2
**Rating:** 4
**Confidence:** 5

**Summary:**

With identifying the property that low-frequency features are primarily recovered in the early stages in the denoising process of diffusion models, this paper proposed a meta-learning method for PTQ in diffusion models, which jointly optimizes the contributions of calibration samples and the weighting of frequency components at each time step. Experimental results reveal the effectiveness of the proposed PTQ method for diffusion models.

**Strengths:**

1. The proposed method well-solved the motivations claimed by authors.
2. The formulation of equations are clear.

**Weaknesses:**

1. For measuring the impact of frequency on model performance, authors propose a learnable frequency weight. I'm curious about simply apply a set hyper-parameter frequency weight, for example a weight ranges from [0, 1]. Will this simple method also achieves performance improvement? Since the learnable weights may involve quantization time cost, authors should compare their methods with such a naive weights to find a better trade-off between performance and quantization efficiency.
2. It would be better to plot some figures (which are sampled from the calibration dataset) for better explaining the frequency motivation in the section of introduction.

**Questions:**

See weaknesses. I would raise my score if my concerns are solved.

---

> ### Author Response · Authors · 2025-11-21
>
> Dear Reviewer gXhQ,
>
> We greatly appreciate the time and effort the Reviewer dedicated to considering our paper. Here are our responses to all concerns raised by the Reviewer.
>
> ### Weaknesses and Questions
>
> > **W1. Trade-off between learned frequency weighting vs. Simplified version of the method (e.g., static frequency weighting based on a heuristic).**
>
>
> **Answer:** Regarding the trade-off between learned frequency weighting and simpler heuristic approaches, we have conducted experiments to directly address this concern.
>
> We explored a simplified version using a static weighting heuristic for frequency guidance by linearly increasing emphasis on high-frequency subbands and decreasing emphasis on low-frequency subbands over later denoising timesteps. This heuristic approach completely avoids the optimization loop over $\lambda$, reducing overall overhead.
>
> As shown in the Table below for LSUN-Bedrooms W4A8 setting, the static heuristic alone achieves a 0.25 FID improvement over the baseline TFMQ-DM. However, incorporating our learned frequency-weighting scheme further boosts performance to a 0.40 FID improvement. This indicates that our learned frequency weighting can capture the evolution between frequency bands more effectively than a simple heuristic approach.
>
> | Method | FID ↓ |
> |--------|-------|
> | TFMQ-DM (baseline) | 3.68 |
> | + Static frequency heuristic | 3.43 |
> | + Learned frequency weighting (Ours) | 3.28 |
>
> **Table**: Comparison of FID and sFID scores for TFMQ-DM baseline, static frequency heuristic, and our learned frequency weighting on LSUN-Bedrooms W4A8 using LDM-4.
>
>
> > **W2. Visualization the calibration dataset**
>
> **Answer:** We thank the Reviewer for this suggestion. We have added visualizations sampled from the calibration dataset to Appendix A.6 to better explain the motivation for our frequency-based approach. These visualizations provide intuitive evidence supporting this approach and strengthen the motivation.

---

### Author Response · Authors · 2025-12-04
**Summary of the discussion**

Dear Area Chair,

Thank you for taking the time to review our paper and coordinate the rebuttal process. We would like to briefly summarize the discussion and our rebuttal efforts.


### Summary of the discussion

- We sincerely thank all Reviewers for their thoughtful and constructive feedback. We are encouraged that the Reviewers acknowledged several strengths of our work including the proposed method is **well-motivated with clear formulation** (Reviewer gXhQ), the method has **good originality and is worthy of further research and analysis** (Reviewer ruiM), and **strong empirical results** (Reviewer RZMH).

- We provided complete rebuttals to all questions from all three Reviewers. The key concerns raised were: the trade-off between learned against heuristic frequency weighting, comparisons with recent methods, implementation complexity, formal justification for optimality of the sampling distribution, and generalization to other modalities.

- We addressed all concerns raised by the Reviewers: provided ablation studies comparing static against learned frequency weighting, added new experimental comparisons with recent PTQ methods, demonstrated minimal implementation overhead with scalability analysis, provided theoretical grounding for convergence and optimality, and discussed generalization to other modalities such as video and audio.

- Among all the Reviewers, Reviewer ruiM has been supportive of the paper from the beginning with a confidence score of 5. Reviewer gXhQ also indicated a willingness to raise their score if concerns are addressed, with a confidence of 5. Reviewer RZMH provided a lower confidence score of 3, but the Reviewer acknowledged our strong empirical results and the main concerns of the Reviewer are possible future extensions of the current method in theoretical aspect and practicality to other modalities.
$~$
------------------
$~$
### Detailed Discussion

**1. Reviewer gXhQ (Rating: 4, Confidence: 5)**

Reviewer gXhQ asked if a simple static heuristic could replace our learned frequency weighting. The Reviewer also mentioned that "I would raise my score if my concerns are solved". We believe we have fully addressed this concern by conducting ablation studies comparing static frequency heuristics against our learned approach. The results show that while a static heuristic improves over the baseline, our learned approach achieves even better FID improvement. This confirms that learning the frequency evolution is crucial for better performance. We also added calibration dataset visualizations to Appendix A.6 as requested.

**2. Reviewer ruiM (Rating: 6, Confidence: 5)**

Reviewer ruiM requested comparisons with recent PTQ methods (APQ-DM, SVDQuant) and efficiency metrics. We added APQ-DM and TCAQ-DM (a more advanced PTQ method from AAAI 2025) comparisons to Tables 1 and 2, showing our method consistently outperforms these baselines. Regarding SVDQuant, we clarified that our method is orthogonal to SVDQuant. The SVDQuant improves the quantization scheme, while we improve from a data perspective and could complement SVDQuant. We also provided memory/speed metrics showing identical efficiency to TFMQ-DM with better generation quality.

**3. Reviewer RZMH (Rating: 4, Confidence: 3)**

Reviewer RZMH expressed concerns about implementation overhead, formal justification for optimality of the sampling distribution, and generalization to other modalities. We show that our method introduces negligible overhead for performance improvements. We also provided theoretical grounding based on bi-level meta-learning convergence. While global optimality is intractable in nonconvex settings, our approach converges to a local optimum with consistent empirical improvements. Finally, we explained how our sample-weighting strategy is domain-agnostic, allowing generalization to video and audio modalities.


$~$

We kindly hope that the Area Chair will consider these responses when making the final decision.

Thank you very much for your time and for coordinating the review process.

Sincerely,

Authors

---

### Meta-Review · Area_Chair_YgrS · 2026-01-07

**Summary:**

This paper proposes a frequency-aware meta-learning framework to optimize post-training quantization (PTQ) for diffusion models by leveraging the observation that different diffusion timesteps correspond to different frequency characteristics. By introducing Discrete Wavelet Transform (DWT)–based frequency decomposition and learned frequency-dependent sampling weights, the method aims to improve generation quality under quantization constraints. Reviewers generally found the core idea to be novel and potentially impactful, and acknowledged the solid technical formulation and empirical validation. The authors’ rebuttal addressed several reviewer questions through additional ablation studies, comparisons with recent PTQ baselines, and clarifications on efficiency and generalization. While some concerns remain regarding theoretical optimality and broader validation, the overall assessment indicates that the key claims are largely supported and that the work represents a meaningful contribution to the PTQ and diffusion model literature.

**Reviewer Concerns:**

Reviewer gXhQ questioned whether a simple static frequency heuristic could replace the learned frequency weighting mechanism. The authors addressed this concern with additional ablation studies, demonstrating that while static heuristics offer improvements over the baseline, the learned weighting consistently yields superior FID gains.

Reviewer ruiM emphasized the need for comparisons with recent PTQ methods and efficiency analysis. In response, the authors added experimental results against APQ-DM and TCAQ-DM, showing consistent performance improvements, and clarified why they did not compared with SVDQuant. Additional memory and speed evaluations indicated that the proposed method achieves improved generation quality without introducing extra inference overhead, partially alleviating concerns about practical applicability.

Reviewer RZMH raised concerns about implementation complexity, the lack of formal guarantees for the optimality of the learned sampling distribution, and the generalization of the approach beyond image generation. The rebuttal clarified that the implementation overhead is negligible.

**Reviewer Scores:**

I think the authors provided a very comprehensive rebuttal, addressed the concerns from different reviewers. The authors conducted additional experiments to address the question about the simple static frequency heuristic from Reviewer gXhQ (Score: 4). The authors also added more baselines in the experiments to address the concerns from Reviewer ruiM (Score: 6). I think both Reviewers gXhQ and ruiM will increase their scores. Not sure if Reviewer RZMH will change the score or not, as the authors cannot provide formal optimality proofs. Overall, I would think the authors did a nice job in the rebuttal and would like to recommend accepting this paper.

---

### Decision · Program_Chairs · 2026-01-26

Accept (Poster)